# Effect of the Current on the Fire Characteristics of Overloaded Polyvinyl Chloride Copper Wires

**DOI:** 10.3390/polym14214766

**Published:** 2022-11-07

**Authors:** Zhe Li, Qingwen Lin, Yang Li, Huifei Lyu, Huaibin Wang, Junli Sun

**Affiliations:** 1Forensic Science Institute, China People’s Police University, Langfang 065000, China; 2School of Safety Science and Engineering, Xi’an University of Science and Technology (XUST), Xi’an 710054, China; 3Shaanxi Key Laboratory of Prevention and Control of Coal Fire, Xi’an University of Science and Technology (XUST), Xi’an 710054, China

**Keywords:** overloaded PVC copper wire, cone calorimeter, time to ignition, fire characteristics

## Abstract

In this study, the fire behavior variation of unenergized polyvinyl chloride (PVC) copper wires subjected to overload with different currents was investigated by a cone calorimeter. Overload currents were selected from 1 times safe-rated current (*I*_e_) to 3.5 times *I*_e_ to obtain tested sample wires. The mass fraction, time to ignition (TTI), heat release rate (HRR), gas emission, and residue were measured. If the current flowing through the wire increased up to 3.5 times *I*_e_, the TTI of this unenergized wire increased drastically and the peak HRR (pHRR) decreased notably so that the flame growing index (FGI) reduced considerably. When the wire carried less than three times *I*_e_, the FGI remained stable. For all overloaded PVC copper wires, the increase in the heat flux resulted in a higher pHRR and a lower burning duration. However, regardless of the external heat flux exposure, the FGI of copper wires overloaded at 3.5 times *I*_e_ was lower than that of copper wires carrying less than other times *I*_e_. Moreover, the consumption of O_2_ and generation of CO_2_ as the heat flux varied were consistent with that of the HRR. Opposed to expectation, the flame propagation of unenergized PVC copper wires would decline in a fire, if the wire has been damaged by overload with some currents.

## 1. Introduction

In electrical systems, wires heat beyond their temperature rating if the wires are too small or the overcurrent protection device malfunctions [1]. According to NFPA 921-2021, this condition is the overload condition [2]. Prolonged overheating of the overloaded cord causes the progressive degradation of insulation such that the pyrolysis characteristics of the insulation coated on the overloaded wire change. Copper conductors coated with polyvinyl chloride (PVC) are widely applied in residential buildings and industrial premises [3]. In numerous fires involving these wires, high-insulation polymer materials propagate fire rapidly to adjacent areas. In the case of electrical faults, such as short circuit, overloading, and overvoltage, wires conduct more current than their ampacity, which results in the deterioration of insulation materials. Because of the prolonged overheating of the overloaded wire, PVC insulation materials may degrade or decompose and change the fire characteristics over this wire. However, there are a lack of studies on the fire hazard change of the overload wire after its insulation materials have been damaged due to prolonged overheating. Therefore, it is necessary and significant for fire protection to investigate the fire behavior variation of unenergized PVC copper wires with different overload currents flowing before.

Fire propagation apparatus and cone calorimetry have been used to investigate the fire characteristics of electrical wires and cables. Studies have focused on the influence of various factors, such as the size of the wire, direction of the wire, metal core, environment, and insulation material, on the flammability of wires and cables with various high-polymer insulations and sheaths. Numerous studies have been conducted on wire and cable fires. For example, the fire performance of several electrical cables was investigated in the FIPEC project. Mcgrattan et al. [4,5] investigated the HRR, ignition, and flame spread of cables in a tray assembly during a fire. Meinier et al. [6] studied the fire performance of halogen-free flame-retardant electrical cables with a cone calorimeter and discussed the influence of two parameters, namely, external heat flux and the spacing between cables. Zhang et al. [7] measured combustion parameters, such as the heat release rate (HRR) and mass loss rate of cross-linked polyethylene (XLPE) cables under three radiation intensities. Hirschler [8] and Barnes [9] studied the combustion behavior of a series of cables based on ISO 5660 and ASTM d5424 [10] using a cone calorimeter. Babrauskas [11] presented a full-scale peak HRR prediction method for cable trays in the SFPE Handbook. Emanuelsson et al. [12] studied the effect of accelerated aging on the fire performance of building wires. Recently, Magalie et al. [13] and Kim et al. [14] used a cone calorimeter to measure the fire performance of small halogen-free cables and nuclear power plant cables, respectively. In particular, Wang et al. [15,16] studied the influence of two key factors, namely, external heat flux and thermal aging, on the fire characteristics of cables and reported that thermal aging could modify the chemical compositions and structures, which increases time to ignition (TTI) and decreases the peak HRR (pHRR) during wire burning. Zhang et al. [17] explored the effect of aging time and type of insulation material on the fire hazard of cables. Compared with the thermal aging of external heating, the hot metal core resulting from overload deteriorates the insulation coating, which may likewise change the fire characteristics of faulted wires. Limited studies have been conducted on the fire characteristics of overloaded PVC copper wires. Thus, the fire performance of wires, which have carried different overload currents before, is also of great interest at present.

In this study, first, the procedure of PVC copper wire overloading is described. Next, the fire behaviors of PVC copper wires overloaded at 1.5, 2, 2.5, 3, and 3.5 times *I*_e_ for one hour are investigated using a cone calorimeter. Mass fraction, TTI, HRR, total heat release (THR), gas emission, and residue were measured under various external heat fluxes. Finally, the flaming growing index (FGI) is used to evaluate various flowing current effects on the fire performance of overloaded PVC copper wires.

## 2. Materials and Experimental Procedure

### 2.1. Materials

In this study, multi-stranded copper conductors coated with PVC, termed PVC copper wires, supplied by Chint Group Co., Ltd. in Taizhou, Zhejiang Province, China, were used. The core of these wires, including 19 copper strands, have an external diameter of 3.39 mm and a thickness of 0.8 mm and are widely applied in residential and commercial buildings. According to Chinese Standard GB/T 19666-2019 [18] and IEC 60331-2 [19], the safe-rate current (*I*_e_) of this sample wire was 20 A. Multi-stranded PVC copper wire specification parameters are shown in Table 1.

### 2.2. Experimental

#### 2.2.1. Overloading Procedure

The sample wire with a length of 1 m was connected to the electrical fault simulation device shown in Figure 1. The device was used to energize the sample wire. This electrical fault simulation device can be adjusted to allow various VAC constant currents to flow through the horizontal sample wire, including 30, 40, 50, 60, and 70 A. The relationship between the current value (A) and the current times (*I*_e_) is presented in Table 2. Each current was sustained and allowed to flow through the wire for one hour so that the wire insulation had sufficient time to degrade. In actual application, PVC insulation visibly alters the integrity when the wire carries current at more than 3.5 *I*_e_, so that the faulty wire can be replaced. By contrast, wires overloaded with less than 3.5 *I*_e_ cannot be observed, so the faulted wires will not be found and substituted. Therefore, overload currents were in the range of less than 3.5 *I*_e_ to prepare sample wires tested by cone calorimetry. Figure 2 shows the appearances of PVC copper wires carrying various overload currents. The insulation color of 3.5 *I*_e_ PVC copper wires can be distinguished from that of other sample wires, which indicates that some composition and structural changes may occur. Subsequently, the wire sample in the sealing bag was under constant conditions until the fire characteristic could be measured and evaluated through cone calorimetry.

#### 2.2.2. Cone Calorimeter

The fire characteristics of unenergized sample wires was measured and examined by a cone calorimeter from TESTech Instrument Technologies Co., Ltd., Suzhou, Jiangsu Province, China. The cone calorimeter is a small-scale calorimeter that is widely used to investigate fire characteristics of solid combustibles. Numerous flammability parameters, such as TTI, HRR, mass loss, and emission gases, can be obtained using this cone calorimeter method based on oxygen consumption. For the fire characteristic of electric wires and cables, this method is reliable and exhibits excellent reproducibility and repeatability. Studies have been conducted to evaluate the fire characteristics of electric wires [3,11], and the bench-scale results from a cone calorimeter have been verified to correlate with the results from large-scale fire tests. The experiments were based on the ISO 5660 standard [10], and the sample wire was cut into pieces with a length of 10 cm. As shown in Figure 3, 25 pieces of sample wires were placed in parallel on the sample holder (10 cm × 10 cm × 10 cm). Thus, no space existed between each piece. The horizontal surface holding the sample wire was elevated nearly up to the top edge by mineral wool in the sample holder and coated with silver paper to ensure that all sample wires could be uniformly heated. External heat fluxes of 25, 30, 35, 40, and 50 kW/m^2^, which are typically encountered in fires, were tested. To ensure the accuracy of the experiments, two parallel experiments were performed for each group of experiments.

## 3. Results and Discussion

### 3.1. Ignition of Wires

The ignition of solid materials is critical for fire spread and growth. The TTI is a critical parameter for evaluating the difficulty degree for some solid materials to be ignited. The TTI results of wires overloaded at 1.5, 2, 2.5, 3, and 3.5 *I*_e_ are shown in Figure 4 and Figure 5. The TTI is inversely related to the external heat flux. The average TTIs corresponding to 1.5, 2, 2.5, 3, and 3.5 *I*_e_ were 55, 56, 54, 52, and 59 s, respectively, at the heat flux of 25 kW/m^2^. When the external heat flux increased to 50 kW m^−2^, the average TTI decreased by 17, 17, 18, 18, and 20 s, respectively. Thus, the external heat flux intensity considerably influenced the ignition characteristics of PVC insulation. At high radiation intensities, the thermal decomposition of PVC was accelerated. After the material was heated by the external heat flux, it decomposed to produce combustible gases, such as H_2_, CH_4_, and C_2_H_4_ [20], which were ignited by the intermittent arcing and eventually generated a continuous flame. These results illustrate that wires overloaded at 3.5 *I*_e_ exhibit higher TTI values than other wires overloaded at lower current values. Under the same heat conditions, the TTI value of the wire subject to a higher overload value before is greater than that of the wire with a smaller overload value because the active composition and structure of the PVC insulating layer undergo chemical changes under Joule heating from the copper core when the wire is overloaded. The TTI variation of PVC copper wires induced by overload is analogous to that of thermal aging PVC copper wires. According to previous studies [16], the increase of TTI is attributed to chain breakage, HCl emission, and plasticizer volatility or decomposition, which lead to elevated hardness and brittleness.

Calculation models have been developed to predict the TTI of flat sheets under various external heat fluxes using a cone calorimeter, as expressed in Equation (1), and related to thermally thick and thin samples [19,20,21]:(1)$$TTI=\{\begin{array}{c} \frac{\pi }{4}k\rho c{\left(\frac{T_{ig}-T_{\infty }}{{\dot{q}}_{e}^{\prime \prime }-{\dot{q}}_{ig,crit}^{\prime \prime }}\right)}^{2}\\ \rho cd\left(\frac{T_{ig}-T_{\infty }}{{\dot{q}}_{e}^{\prime \prime }-{\dot{q}}_{ig,crit}^{\prime \prime }}\right) \end{array}$$
where *k*, *ρ*, *c*, and *d* are the thermal conductivity (W m^−1^ K^−1^), density (kg m^−3^), specific heat (J kg^−3^ K^−1^), and sample thickness (m), respectively. Here, $T_{ig}$, $T_{\infty }$, ${\dot{q}}_{e}^{\prime \prime }$, and ${\dot{q}}_{ig,crit}^{\prime \prime }$ represent the temperature to ignition (K), room temperature (K), the external heat flux (kW m^−2^), and the critical heat flux. If *k*, *ρ*, *c*, $T_{ig}$, $T_{\infty }$, and ${\dot{q}}_{ig,crit}^{\prime \prime }$ are constant, then the natural logarithm of TTI is linearly related to the natural logarithm of ${\dot{q}}_{e}^{\prime \prime }$, as follows:(2)$$TTI\propto {\dot{q}}_{e}^{\prime \prime }{}^{n}$$

When the index *n* is higher, igniting the sample becomes difficult. Index 2 is critical for distinguishing the thermal thick sample from the thin sample [20,21]. The linear relationship between ln(TTI) and $\ln\left({\dot{q}}_{e}^{\prime \prime }\right)$ is presented in Figure 6. Index *n* is between 1.19 and 1.72, which is a thermal thin material. Because of the minimum value of the wire overloaded three times *I*_e_ at 25 kW m^−2^, index *n* is the minimum among all wires overloaded at various currents. The TTI values depend on the flammability properties of insulation materials, which imply chemical–physical change during the internal heating of the thermal core. Because of the similar index and the nearly coincident line of wires overloaded with 1.5, 2, and 2.5 *I*_e_, the physical–chemical properties exhibit considerable alteration.

### 3.2. Heat Release Rate

The HRR is the most essential variable in evaluating the fire hazard of combustible materials. The HRR of organic solid materials plays a key role in the fire behavior and should be considered. The first process of wire combustion is that the insulation is gradually heated and decomposed to produce flammable gases. Based on Thornton’s rule [22], the oxygen consumed in a combustion system can determine the net heat released. To date, the oxygen consumption method is as the most valid method for estimating the HRR of organic solids without considering their combustion chemical [23]. The HRR can be calculated using the following:(3)$$HRR=\frac{E\left({\dot{m}}_{o_{2}}^{0}-{\dot{m}}_{o_{2}}^{e}\right)}{A_{s}}$$
where ${\dot{m}}_{o_{2}}^{0}$, ${\dot{m}}_{o_{2}}^{e}$, *E*, and $A_{s}$ are the mass flow rates of oxygen from the incoming air, the mass flow rates of oxygen from the exhaust duct, the energy released per mass unit of oxygen consumed (~13.1 ± 5% MJ kg^−1^), and the initial exposed surface area (m^2^), respectively. The cone calorimeter is used to evaluate the HRRs of various overloaded wires based on the oxygen consumption principle.

Figure 7 illustrates the HRR curves of overloaded wires carrying various currents exposed to various external heat fluxes. The outer surface layer of PVC insulation is the first component of decomposition. When the wire is ignited, the HRR increases immediately to a peak (pHRR) and subsequently decreases progressively over a few hundred seconds. For the overloaded wire, the thermal degradation of PVC insulation occurs from the inside to outside because of the heating of the internal copper core. Therefore, when the current flowing in the wire is low, then sufficient Joule heat cannot be generated to initiate the chemical–physical change of the outer surface layer. Figure 8 shows the pHRR curves of wires subjected to different overload currents. With an increase in the heat flux, PVC polymer decomposition is accelerated, which results in an increase in the HRR value and an increase in pHRR. As indicated in the pHRR curves of 1.5, 2, 2.5, and 3 *I*_e_, these wires damaged by various currents require nearly the same time to reach at pHRRs, which are approximately identical. Similar to the thermal aging PVC copper wire [7,15], the wire overloaded at 3.5 *I*_e_ requires more time to increase its pHRR, which is lower than that of other overloaded wires exposed to various external heat fluxes. This result likewise indicates the 3.5 *I_e_* flowing through the PVC copper wires can deteriorate the active composition and structure of polymer insulation materials.

The pHRR of wire overloaded at 3.5 times *I*_e_ becomes closer to the pHRR of other overloaded wires with the increase in the external heat flux. The time to pHRR tends to decrease and gradually equals the elevation of the external heat flux.

The flame growth index (FGI) [24,25,26] is used to evaluate flame spread and size on solid materials. The FGI is the ratio of pHRR to time to pHRR [27] as follows:(4)$$FGI=\frac{pHRR}{t_{pHRR}}$$
where $t_{pHRR}$ is the time to pHRR (s). Thus, the unit of FGI is kW m^−2^ s. Table 3 presents the specific values of the pHRR, $t_{pHRR}$, and FGI in two parallel tests. Figure 9 presents the profile of FGIs of various overloaded wires exposed to various external heat fluxes. The FGI of 3.5 *I*_e_ overloaded wires is constantly less than that of other overloaded wires. The FGI linear lines of wires overloaded with 1.5, 2, 2.5, and 3 *I*_e_ are nearly coincident under various external heat fluxes. This result illustrates that the PVC wire carrying less than three times *I*_e_ for one hour cannot accelerate the flame spread through the wire, but the flame spread and size may decrease with the overcurrent elevated continuously. The FGI of 3.5 *I*_e_ overloaded wires decreases considerably, but the FGI of wires with less than three times *I*_e_ overloaded wire is stable. Like the results of previous study [14], the FGI of copper wires decreased due to more deterioration of polymer insulation materials, as the exposure intensity of the heat source increased. This result can be attributed to an increase in the currents, which increases the temperature of the copper core, deteriorates the composition and structure of PVC insulation, and reduces the fire hazard of PVC copper wires.

### 3.3. Gas Emissions

During burning, oxygen (O_2_) [28] in the ambient air is consumed and gaseous combustion products are released. Major species concentrations and emission yields were continuously tracked and measured using a Ttech-16172 Gas Analyzer. The concentration and production of carbon monoxide (CO) [29,30] are low and difficult to observe and therefore ignored. Figure 10 shows the evolution of gas emissions during external flux testing in the range of 25 to 50 kW m^−2^. The consumption of O_2_ exhibited a trend of first decreasing and subsequently increasing with the increase of time, and the production of CO_2_ [31] exhibited a trend of first increasing and subsequently decreasing. The time to reach the peak is similar to the time to reach pHRR. The larger the external heat flux is, the greater the amount of CO_2_ produced by the solid carbon is. The overload current also affects the consumption of O_2_ and the production of CO_2_. When the external heat flux was 35 and 40 kW m^−2^, the consumption of O_2_ and the generation of CO_2_ of wire overloaded with 1.5 *I*_e_ reached the maximum, which indicated the combustion was most sufficient. For the overload current, the trend of the CO_2_ generation was similar to that of HRR. Furthermore, the generation of CO_2_ from 3.5 *I*_e_ overloaded wires was the minimum among all tested wires regardless of various heat fluxes. As the heat flux increased, the value of CO_2_ generation of 3.5 *I*_e_ overloaded wire tended to equal those of other overloaded wires.

### 3.4. Residues and Mass Loss

After the burning stopped, a large amount of residue remained from the PVC wires. The mass loss of residue was between 16.8 and 21.6 g, as shown in Figure 11. With the external heat flux increasing, the mass loss increased. The mass of residues left for 3 *I*_e_ overloaded wire at 25 kW m^−2^ was the largest among all tested overloaded wires, which is consistent with the results analyzed in Section 3.1. Under all external heat fluxes [32,33], the mass loss fraction of 3.5 *I*_e_ overloaded wire stably remained the least compared with that of other overloaded wires. Besides the 3.5 *I*_e_ overloaded wire, the mass loss fractions of other overloaded wire were close and simultaneously become closer as the heat flux elevated. Figure 12 shows images of burning residues under various heat fluxes. These residues are similar but differ slightly in appearance. For wires overloaded with 1.5, 2, 2.5, and 3 *I*_e_, the lines of each wire in residues cannot be observed, but the lines in 3.5 *I*_e_ overloaded wire residues are visible. This result indicates that the intensity of 3.5 *I*_e_ overloaded wire combustion is less than that of other overloaded wires.

## 4. Conclusions

To understand the fire hazard change of wires resulting from the deterioration of PVC insulation materials induced by Joule heating of the overcurrent core, PVC copper wires were used to investigate the effect of various overload currents on fire propagation factors and behaviors over wires subjected to overload currents before using a cone calorimeter. The main conclusions are as follows.

(1) Unless the structural integrity of insulation is damaged visibly for overloaded PVC copper wires, wires overloaded at less than 3 times *I*_e_ have much the same TTI. If 3.5 times *I_e_* flows through the wire before a fire, this overloaded wire would exhibit higher TTI values regardless of the heat flux. For this size PVC copper wire, the critical overload current, which remarkably alters the flame spread and growth ability over the wire, ranges from 60 to 70 A.

(2) For wires overloaded at 1.5, 2, 2.5, and 3 times *I*_e_ and exposed to various heat fluxes, the pHRRs and the times to pHRR were similar; however, the wire overloaded at 3.5 times *I*_e_ exhibited lower pHRR and required more time. By calculating the FGI of wire overloaded at each *I*_e_, the increase in overload current up to 3.5 *I*_e_ could decrease the flame spread and size over PVC copper wires overloaded before.

(3) Considering the effect of overload currents, the trend of the CO_2_ generation is similar to that of the HRR, which originates from the generation of CO_2_ from wires overloaded at 3.5 times *I*_e_ and is the minimum among all tested wires. As the heat flux increased, the value of CO_2_ generation of 3.5 *I*_e_ overloaded wire tended to equal those of other overloaded wires. In contrast to other wires subjected to overload at less than 3 times *I_e_*, the residue in wires overloaded at 3.5 times *I*_e_ indicates that its combustion intensity declined considerably.

In summary, the flame intensity and propagation of PVC copper wire would decline, if the wire was damaged by overload of 3.5 times *I_e_* before a fire, although the structural integrity of insulation cannot be visibly observed. Thus, to some extent, the increase in the overload current reduced flame propagation in faulted PVC copper wire. However, further studies are required to investigate the effect of overload currents on the electrical insulating property of polymer insulation materials to comprehensively investigate the fire hazard change.

## Figures and Tables

**Figure 1 polymers-14-04766-f001:**
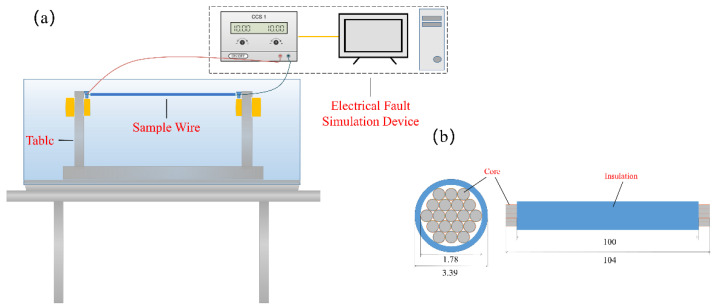
Schematic of the experimental apparatus: (**a**) electrical fault simulation device and (**b**) sample wire.

**Figure 2 polymers-14-04766-f002:**
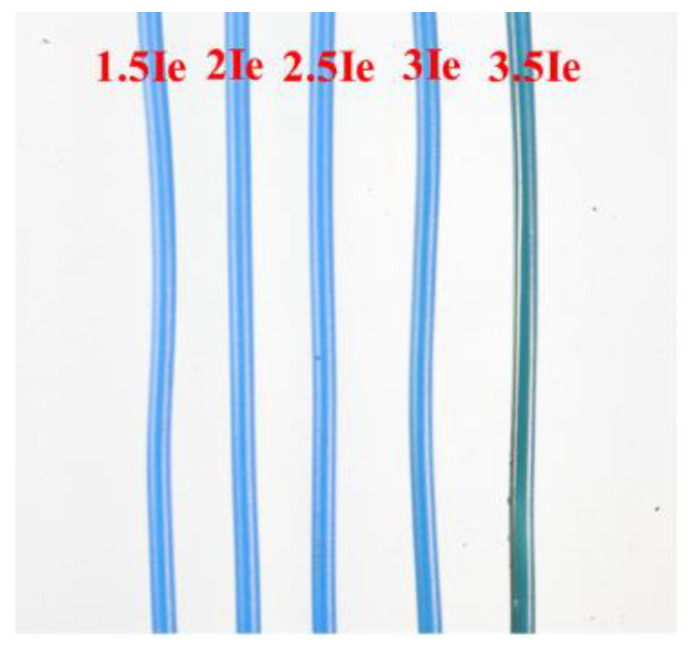
Various overload current PVC copper wires.

**Figure 3 polymers-14-04766-f003:**

Position of the overloaded PVC copper wires in the sample holder.

**Figure 4 polymers-14-04766-f004:**
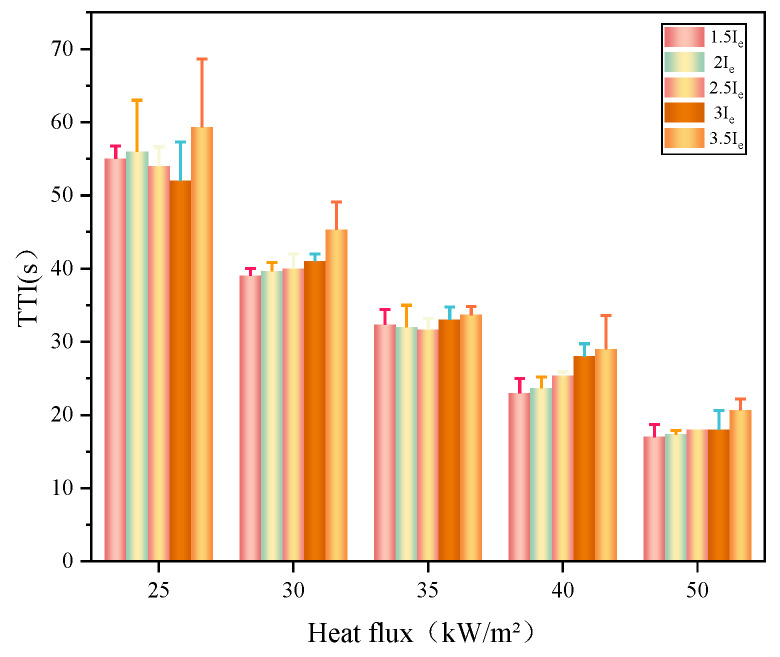
TTI values for overloaded wires.

**Figure 5 polymers-14-04766-f005:**
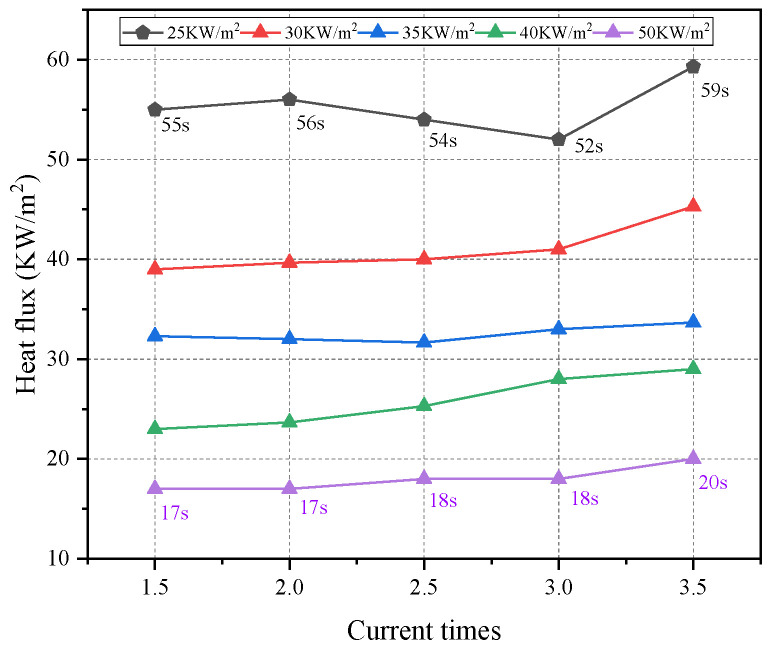
TTI under various external heat flux and current times.

**Figure 6 polymers-14-04766-f006:**
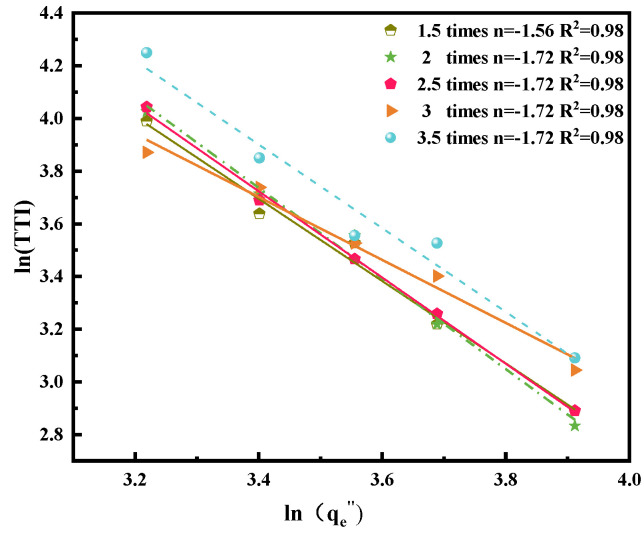
ln(TTI) and $\ln\left({\dot{q}}_{e}^{\prime \prime }\right)$ fitted curve.

**Figure 7 polymers-14-04766-f007:**
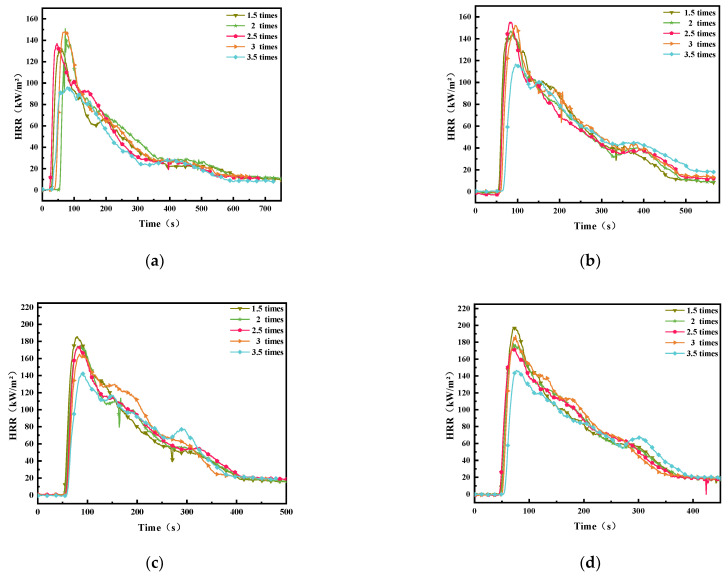
HRR for the wires at various external heat fluxes. (**a**): 25 kW m^−2^; (**b**): 30 kW m^−2^; (**c**): 35 kW m^−2^; (**d**): 40 kW m^−2^; (**e**): 50 kW m^−2^.

**Figure 8 polymers-14-04766-f008:**
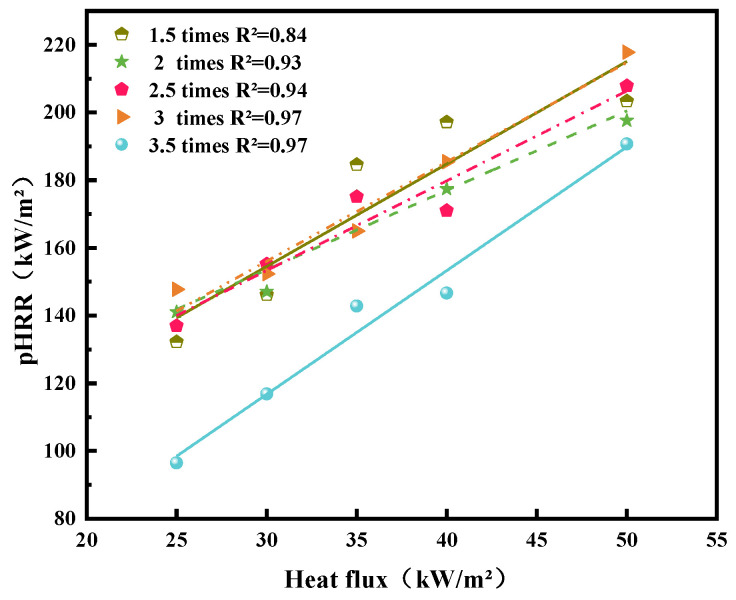
pHRR for the wires at various external heat fluxes.

**Figure 9 polymers-14-04766-f009:**
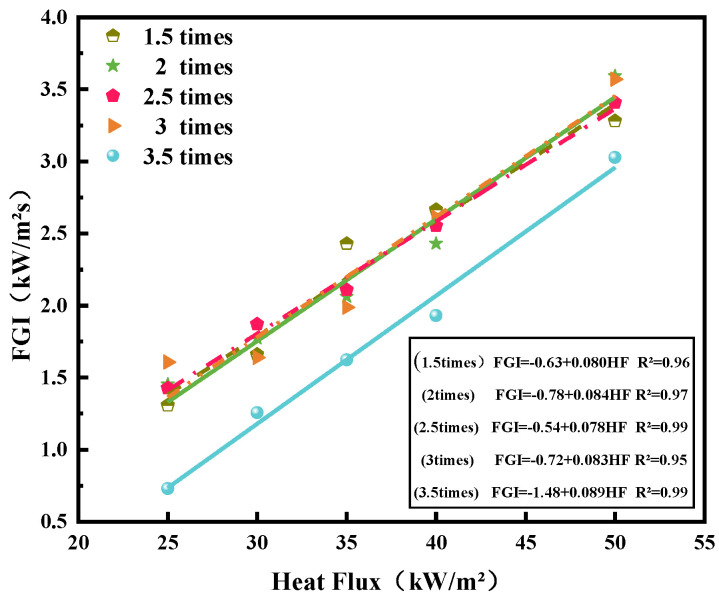
FGI for the wires at various external heat fluxes.

**Figure 10 polymers-14-04766-f010:**
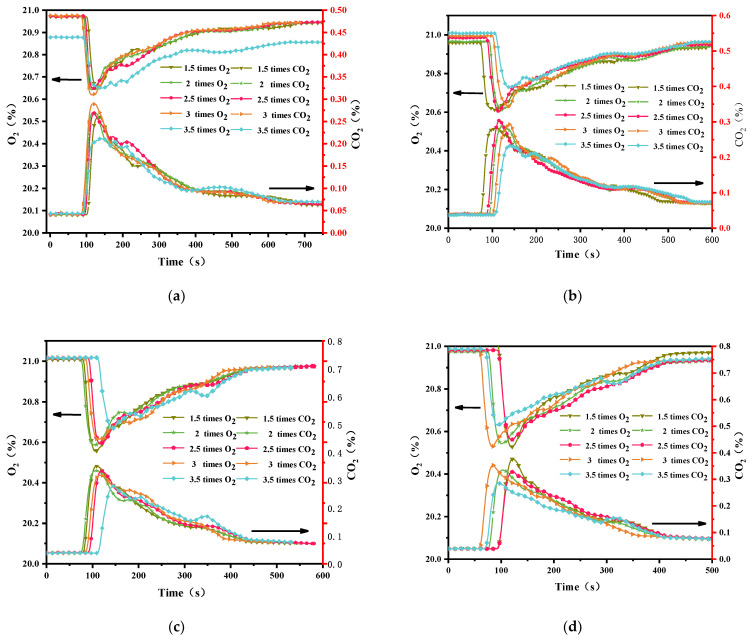
Evolution of the gaseous emissions at various heat fluxes. (**a**): 25 kW m^−2^; (**b**): 30 kW m^−2^; (**c**): 35 kW m^−2^; (**d**): 40 kW m^−2^; (**e**): 50 kW m^−2^.

**Figure 11 polymers-14-04766-f011:**
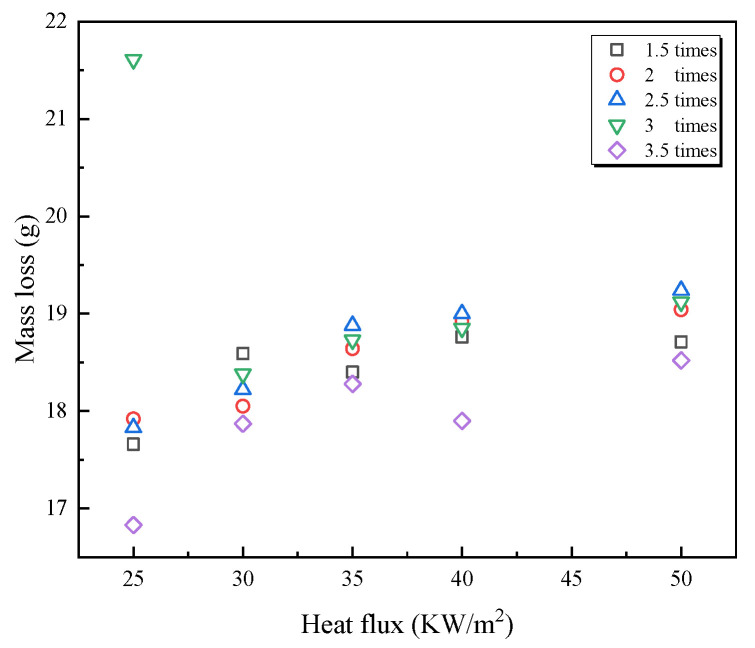
Mass loss under various heat flux and overload current conditions.

**Figure 12 polymers-14-04766-f012:**
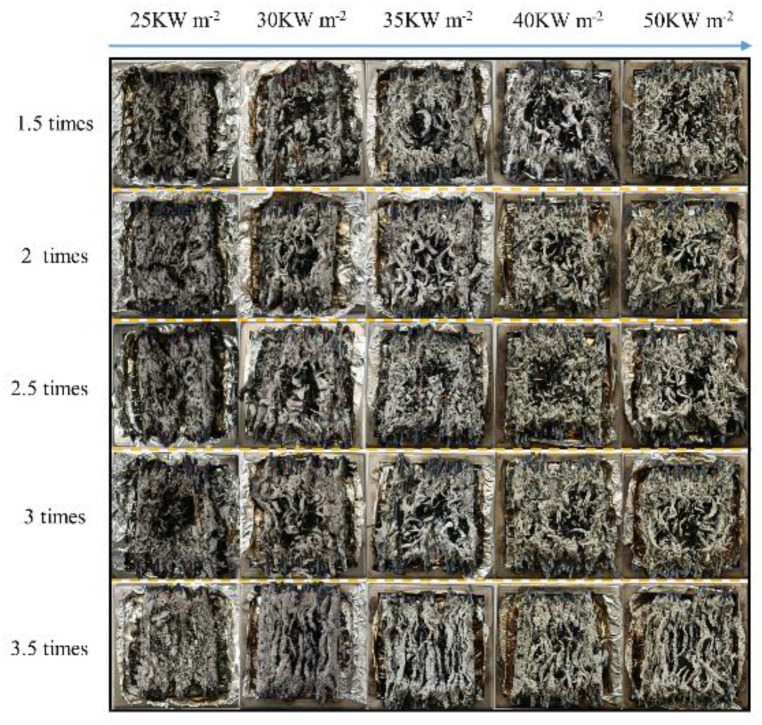
Image of the burning residue at various heat fluxes.

**Table 1 polymers-14-04766-t001:** Multi-stranded PVC copper wire specification parameters.

Multi-Stranded PVC Copper Wire	
Model	BVR
Number of core	19
Inner diameter	3.3 mm
Outside diameter	4.1 mm
Cross-sectional area	2.5 mm^2^
Safe rate current	20 A

**Table 2 polymers-14-04766-t002:** Relationship between current value (A) and current times (*I*_e_).

Number	Current Times (*I*_e_)	Current Value (A)
1	1.5	30
2	2	40
3	2.5	50
4	3	60
5	3.5	70

**Table 3 polymers-14-04766-t003:** pHRR, $t_{pHRR}$, and FGI in two parallel tests.

		Current Times (*I*_e_)														
Heat Flux (kW/m^2^)		1.5			2			2.5			3			3.5		
Number		1	2	average	1	2	average	1	2	average	1	2	average	1	2	average
pHRR (kW/m^2^)	25	132.2	134.9	133.5	141	140.	140.6	136.9	149.8	143.	147.7	150.9	149.3	96.4	85.1	90.8
	30	146.1	144.2	145.1	147.0	139.5	143.2	155.2	124.5	139.8	152.3	132.3	142.3	116.8	116.8	116.8
	35	184.5	156.1	170.3	175.1	152.2	163.6	175.0	149.2	162.1	164.9	144.1	154.5	142.8	129.4	136.1
	40	197.1	169.8	183.5	177.4	147.6	162.5	170.9	156.5	163.7	185.4	171.8	178.6	146.6	127.1	136.9
	50	203.3	173.8	188.6	197.5	197.6	197.5	207.8	192.1	200.0	217.7	189.7	203.7	190.6	150.9	170.8
t-pHRR (s)	25	101	99	100	97	103	100	96	99	97	92	97	94	132	152	142
	30	88	73	80	83	83	83	83	106	94	93	93	93	93	106	99
	35	76	68	72	85	68	76	83	69	76	83	82	82	88	80	84
	40	74	62	68	73	83	78	67	65	66	71	67	69	76	71	73
	50	62	51	56	55	53	54	61	52	56	61	48	54	63	61	62
FGI (kW/m^2^)	25	1.30	1.36	1.33	1.45	1.36	1.40	1.42	1.51	1.47	1.60	1.55	1.58	0.73	0.56	0.64
	30	1.66	1.97	1.81	1.77	1.68	1.72	1.87	1.17	1.52	1.63	1.42	1.53	1.25	1.10	1.17
	35	2.42	2.29	2.36	2.06	2.23	2.14	2.10	2.16	2.13	1.98	1.75	1.87	1.62	1.61	1.62
	40	2.66	2.73	2.70	2.43	1.77	2.10	2.55	2.40	2.48	2.61	2.56	2.58	1.92	1.79	1.86
	50	3.28	3.40	3.34	3.59	3.72	3.66	3.40	3.69	3.55	3.56	3.95	3.76	3.02	2.47	2.75

## Data Availability

The data presented in this study are available on request from the corresponding author.

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
