# Peer review of "Effect of the Current on the Fire Characteristics of Overloaded Polyvinyl Chloride Copper Wires"

_polymers, 2022, doi:10.3390/polym14214766_

Round 1

Reviewer 1 Report

My comments are following:

1. authors need to revise the manuscript' title.

2. The main aims of this study should be highlighted in abstract section.

3. The literature flow not clear, more recent studies in this area should discuss and cite. As well as, authors should focus to highlight the novelty of this study.

4. Manuscript not well organized, for example, in methodology section more details about materials and tests should include.

5. For each finding, authors need to provide the reason with more appropriate discussion as well as compare to literature.

6. The quality of most the figures should be improve.

7. Authors should modify the conclusion section and highlight the important findings and achieved aims. 

Reviewer 2 Report

Thanks for the effort in writing the following paper. Please consider the following comments: 

 - You may want to rephrase the title as it has repeating words (overloaded)

- Abstract: How is it possible that the increase in current causes decreases in heat release rate? Please explain if I am wrong 

     If the current flowing through the wire increased up to 3.5 times Ie for one hour, the TTI increased drastically, and the peak HRR (pHRR) decreased considerably ...

- Line: 60 : You claim " Therefore, the effect of overload currents on the fire behaviors of faulty wires is yet to be investigated"  Is the paper is about faulty wires, you need to emphasize it in the abstract and title. Also, please be explicit about what parameters your study includes that the literature doesn't have. Can you make a table to compare the parameters what is covered in literature and shows clearly your contribution 

-Line: 74: What is the name of the Chinese Standard/Compliance document? 

- do you have a picture of the experimental setup? Please put the picture of the setup. Do you have calibration data of the Cone calorimeter by any chance? 

- figure 8. tHHR? or pHHR?

Round 2

Reviewer 1 Report

Authors modified the manuscript content.